# The relationship between age and sex partner counts during the mpox outbreak in the UK, 2022

**Julii Brainard**[1], **Louise E. Smith**[2]\*, **Henry W. W. Potts**[3], **G. James Rubin**[2]

**1** Norwich Medical School, University of East Anglia Norwich, Norwich, United Kingdom, **2** Institute of Psychiatry, Psychology and Neuroscience, King's College London, London, United Kingdom, **3** Institute of Health Informatics, University College London, London, United Kingdom

\* louise.e.smith@kcl.ac.uk

**Data Availability Statement:** All relevant data are located at https://osf.io/ad7yj/.

**Funding:** The study was funded by the National Institute for Health and Care Research (NIHR). This work was funded by the National Institute for

## Abstract

### Background

Understanding the dynamics of an infectious disease outbreak linked to sexual activity requires valid expectations of likely counts of unique sex partners during the infectious period. Typically, age is the key demographic trait linked to expected partner count, with many transmission models removing adults from the sexually active pool abruptly at a pre-specified age threshold. Modelling the rate of decline in partner counts with age would benefit from a better description of empirical evidence.

### Methods

During the 2022 mpox epidemic in the UK, we asked individuals about their partner counts in the preceding three weeks, which is about the same as usual infectious period for persons with active mpox. We used negative binomial regression (all responses) and Weibull regression (non-zero responses) to analyse the relationship between age and partner counts, adjusted for other demographic data (such as education level and occupation), sub-dividing by three types of respondent: men who have sex with men (MSM), men who have sex with women, and women who have sex with men.

### Results

Most respondents had zero or one recent partner, all distributions were skewed. There was a relatively linear declining relationship between age and partner counts for heterosexual partnership groups, but a peak in partner counts and concurrency for MSMs in middle age years (age 35–54), especially for MSM who seemed to be in a highly sexually active subgroup.

### Conclusion

Useful data were collected that can be used to describe sex partner counts during the British mpox epidemic and that show distinctive partner count relationships with age, dependent on partnership type.

Health and Care Research Health Protection
Research Unit (NIHR HPRU) in Emergency
Preparedness and Response, a partnership
between the UK Health Security Agency (UKHSA),
King's College London (KCL) and the University of
East Anglia (UEA). The views expressed are those
of the authors and not necessarily those of the
NIHR, UKHSA, Department of Health and Social
Care, UEA, KCL, or University College London
(UCL). For the purpose of open access, the author
has applied a Creative Commons Attribution (CC
BY) licence to any Author Accepted Manuscript
version arising. The funders had no role in study
design, data collection and analysis, decision to
publish, or preparation of the manuscript.

**Competing interests:** HWWP has a PhD student
who works at and has fees paid by AstraZeneca.
This does not alter our adherence to PLOS ONE
policies on sharing data and materials. All other
authors declare that we have no conflict of interest.

## Introduction

Human mpox is a smallpox-like zoonotic infection caused by a virus in the Orthopoxvirus genus. Outbreaks in multiple countries and continents was identified in 2022 [1], especially in high income countries, almost exclusively in men who have sex with men (MSM). Most 2022 cases were linked to sexual activity within this community. Understanding the dynamics of an infectious disease outbreak linked to sexual activity can benefit from collecting data about social contact patterns, specifically counts of unique sex partners (which for simplicity, we refer to simply as 'partners' throughout this report). It is known that high partner counts for just a small minority of individuals are important drivers in sustaining and spreading diseases that have intimate contact as an important transmission pathway [2]. Hence, reliable description of the distribution of unique partner counts during infectious periods is a key parameter for establishing credibility in disease transmission models dependent on intimate contact. Moreover, models that try to evaluate the merits of different infection control strategies in real-world populations may need to rely on plausible assumptions about distributions of partner counts that are known to relate to commonly measured demographic traits, especially age [3,4]. That published data about the relationship between age and partnership counts could be unrepresentative because of sampling methods was suggested previously: for instance, because of broad under-sampling of MSM [5] or because of dependence on Internet Apps that under-sampled older MSM [6].

This study is a secondary analysis of part of a dataset collected in the UK between 5 September and 6 October 2022. The primary aims of the study that collected those data [7] were to survey the general population about mpox knowledge and undertake a randomised and controlled experiment about health messaging and behavioural intentions. The survey over-sampled MSM and addressed some aspects of sexual history: partner count of unique males or females in the last three weeks and three months separately. This article uses part of the survey data to describe the distribution of partner counts in three types of partnership groups: MSM, women who have sex with men (WSM), and men who have sex with women (MSW). At the time of the 2022 data collection, the most recent similar surveys and analyses were based on information collected by two National Surveys of Sexual Attitudes and Lifestyles (Natsal) and a single Natsal survey wave as part of the gonococcal resistance to antimicrobials surveillance programme [2,8]. Those three Natsal surveys were last updated in 2012 and had (combined) 1017 MSM respondents. A fourth Natsal survey (www.natsal.ac.uk/natsal-survey/natsal-4) started in 2022 and expects to complete data collection in 2023. The Natsal surveys provide rich data about specific sexual habits, including information about partner counts in lifetime or over prior 12 months [9]. In contrast, the original data that we analyse here apply to a narrower time period, one that was especially important in the context of concurrent public health need to understand and contain the British mpox epidemic.

Our research aim was to use the autumn 2022 survey data in a secondary analysis to explore the relationship between age and the count of unique partners, adjusting for available confounders. It was valuable to explore the relationship between age and partner counts because age is a key determinant of expected partner counts in many models of sexually transmitted diseases. Often in such models, there is no variation in expected partner counts after age 16 or 17. Alternatively, partner count may be expected to decline at a similar rate with increased age for all partnership types [10], or individuals beyond a prespecified age are abruptly retired or removed from the population of interest/considered to be at risk, for instance at age 39–40 years [11,12], 65 years [4,13], or 65–67 years [14]. As at 16 September 2022, the median age of confirmed and highly probable cases from the 2022 UK mpox epidemic [15] was 36 years (IQR 30 to 44). About 6% were age 16–25, about 6% were age 55+. Although most cases (70%)

were age 25–44, a large minority of detected UK mpox cases (30%) were outside this central age group and there were fewer cases aged 16–24 than cases age 45+. Our research objectives were to apply statistical models to the autumn 2022 survey data to explore if and how much reported partner counts varied with age for common partnership types, and to see if a sharp decline in partner count at any specific age threshold (as often assumed in existing STI models) was apparent.

## Methods

### Recruitment strategy

The data were drawn from a cross-sectional survey administered to UK adults from 5 September to 6 October 2022 (inclusive) via a market research company (Savanta) and social media platforms. The survey asked about intended behaviour in a factorial design experiment following exposure to information or motivational messages. Detail on the survey development, recruitment and experiment results are reported elsewhere [7]. Ethical approval for this study was given by the King's College London Psychiatry, Nursing and Midwifery Research Ethics Panel (reference number: LRS/DP-21/22-32287). Participants gave informed digital written consent before beginning survey materials.

Savanta has registered panels of approximately 150,000 prospective survey respondents. Savanta recruitment was in two cohorts: UK general population and a boosted extra sample of UK MSM (Savanta panel members who self-identified as male and as gay, bisexual, or having sex with men; termed Savanta B for UKMSM "boost"). The Savanta general population survey employed quota sampling based on specific socio-demographic characteristics (age, gender, socio-economic grade and region) that matched the national profile. The Savanta boost recruitment applied the same quotas except for gender (because all respondents had to be male).

Recruitment to the same survey was also via adverts on the online social networks Grindr, Facebook and Instagram. In practice, social media respondents were samples of convenience. Facebook and Instagram adverts were targeted at users based on their known engagement profile and apparent interests that made it likely they were MSM, as identified by proprietary profiling algorithms used on these platforms. These algorithms are likely to be similar on both platforms, because Facebook and Instagram are owned by the same parent-company. As a result, data from respondents recruited via Facebook and Instagram were pooled and are designated in this report by their parent company name, Meta. Grindr is a dating application for members of the MSM community and, at the time of survey, was well-established as a successful network for finding casual gay sex.

### Partner count questions and subgroup definitions

Sexual contact was defined as any genital contact. All respondents were asked 4 questions about their counts of recent persons with whom they had sexual contact. Counts were in the last 3 weeks and in last 90 days, and asked separately for male and female partners. Participants typed in the number of partners of each gender with an allowed answer range of 0 to 100 (partners in last 3 weeks) or 0 to 400 (partners in last 90 days).

Women who have sex with women and other partnership types were not investigated because of relatively much smaller (<< 1000) respondent counts. MSM, WSM and MSW populations (partnership groups) as analysed here were defined by reported sexual activity with the stated partner type or reported identity, regardless of any dominant preference. Therefore, candidate MSM (likely MSM population for analysis) were persons with male gender identity who reported on how many male partners they had in the last three weeks (zero or greater)

and either had sexual contact with males in the last 3 months or indicated on a separate question that their sexual preference was gay, queer or bisexual. Similarly, MSW were defined as persons with male gender identity who self-identified either as heterosexual or bisexual and gave data (zero or greater) on recent female partner count or who had had sexual contact with any females in the prior 3 months. The definitions for MSM and MSW were not mutually exclusively. A male participant could be both MSM and MSW. WSM were defined as persons with female gender identity who gave a count (zero or greater) of male partners in the last three weeks, and who either self-identified as heterosexual or bisexual or had sexual contact with any males in the prior 3 months.

We did not distinguish between persons based on whether they had the same gender as that assigned to them at birth; it is possible that mpox transmission risks vary with biological sex and possibly specific sex acts, but at the time of our study no data existed to confirm that. Moreover, it seems likely that transmission risks at this time were dominated by lifestyle factors (such as partner finding habits) that were more closely associated with concurrent gender and sexual preference identity than with biology.

These definitions of candidate MSM, WSM and MSW could under-count the true eligible population who belong in each partnership group who answered the survey, as well as omissions due to some respondents who declined to provide partner count. We do not know if missing data was more likely for persons with very low or very high partner counts. Also, we could not count persons who have low frequency of sexual contact (less than every 90 days) with a suggested partner type (M or W) that was outside their dominant preference. This could mean that we under-estimated the proportion of zero partner count persons in each distribution. Whether respondents were sexually active at all is also a consideration. The survey did not ask if people were sexually inactive (by choice or not, either temporarily or chronically); it was therefore possible that we over-counted the individuals of interest in each partnership population by including sexually inactive individuals in each partnership group who still expressed a sexual orientation preference. We addressed this potential problem by excluding respondents with zero counts for a partner type in preceding three weeks in a Weibull regression (see analysis section below).

## Data quality and cleaning

A survey attention check and cognition check question were used to help detect and filter out automated (fake) or low attention respondents. Among respondents who gave the correct answer to the attention check question, manual inspection of free-text responses to four questions did not suggest that any of these were nonsensical, illogical in the UK context or formulaic sounding which could happen because they were generated in bad faith or by artificial intelligence algorithms.

Respondents were given choices of male, female, non-binary or other to describe their gender. For "other", some respondents described their gender as male or female; these were recoded as male/female where indicated by their open-text description (n = 24). Similarly, for sexual preference, some respondents (n = 30) chose "other" and then described themselves as gay, heterosexual or used other descriptors that enabled them to be categorised as candidate MSM, MSW or WSM. Non-binary gender identity persons were not pooled with females or males or otherwise analysed separately here due to their small representation.

Some answer combinations were logically impossible making those responses unreliable. Eight respondents were excluded who had male gender identity that was the same as their gender assigned at birth *and* who also said they were pregnant. Participants who reported that they had more partners in the last 3 weeks than they had in the previous 90 days were

excluded. This affected 39 respondents about their male partners and 18 respondents about their female partners. One person had partner count answers reassigned as missing because they said they had 100 partners of each sex in the last 3 weeks and 400 partners of each sex in the last 90 days; these were the maximum values possible for all questions and seemed more likely to be data entry error or bad faith responses than accurate information.

## Analysis

Analysis was done in Stata/MP v. 17.0. Significance threshold was p < 0.05. Descriptive statistics are reported for MSM, MSW and WSM stratified by sample origin (Savanta / general population) versus social media: Grindr and Meta. Models were constructed to predict partner count by the three main partnership types: MSM, MSW, WSM. The dependent variable was partner count in the preceding three weeks. Partner count appeared to have a non-linear (curving) but declining relationship with age in exploratory data analysis. Age (whole years) therefore had trial expressions in the models as linear or quadratic, as well as transformed to equal reciprocal × population mean (to allow for possible bias in age of those sampled, Stata code is in supporting information). Partner counts were highly skewed for MSM, and somewhat skewed for MSW and WSM (S1 Fig in S1 Appendix; S2-S9 Tables in S1 Appendix): most respondents had 0 or 1 partners in the past three weeks, while a relatively small number of respondents had high numbers of partner counts (5 or more). To handle the skew, we apply two plausible distributions: negative binomial (including zero-partner count individuals) and Weibull (non-zero data only). The negative binomial distribution has been used for previous predictions of MSM partner or sex-act counts [4,11,14,16]. Removing zero-partner count responses (in last 3 weeks) from the dataset may be justified to explicitly separate individuals who are not sexually active. The Weibull distribution does not predict zero partner counts and was recently recommended especially in the context of understanding rapid spread during the 2022 mpox epidemic because of a 'heavy tail': small numbers of individuals with very high partner counts [2]. These model choices (negative binomial and Weibull) were not intended to comprise a definitive exploration, but rather, their contrast was useful as a sensitivity analysis to indicate if there was consistency in the age-partner count relationship in spite of different modelling approaches.

All models to predict relevant partner count were also adjusted by these other demographic correlates:

- education (with or without a university degree)

- occupation (UK standard occupational classifications [17])

- index of deprivation quintile in resident area [18]

- reported difficulty in paying bills: 4 categories from not at all to extremely difficult [19]

- region of UK (12 areas possible)

- ethnicity, 3 categories: white British; white other; or other (any non-white)

- if they had a regular individual partner with whom a sexual relationship might be presumed (e.g., if married or in a civil partnership, vs. no regular partner)

- dependent child in household (with or without).

The model with best expression of the Age variable was chosen on the basis of minimising the Bayesian information criterion because it penalises model complexity and thus tends to prioritise specificity over sensitivity [20]. We reiterate that we were less interested in finding a

definitive model form than in exploring the relationship between age and partner count, stratified by partnership type. Therefore, our testing of different statistical models (negative binomial or Weibull) had the purpose of seeing if the results were consistent (for age and partner count) rather than deciding which model or distribution was optimal. Also, for reasons of brevity and focus, this article addresses only results for partner count relationship with age (not with any other candidate correlates).

Exploratory data analysis suggested that median partner count strongly varied with recruitment samples (general population, Savanta UKMSM boost, Grindr, or Meta) as shown in stratified frequency counts (S2-S4 Tables in S1 Appendix). Therefore, we also adjusted by recruitment sample in the final models. From the models, we generated and display graphically the predicted number of partners and probability of partner counts = 0, 1, $\geq$ 2 (MSW and WSM), and partner counts = 0, 1, $\geq$ 5 (MSM). For reasons of brevity, we do not present data separately on partner counts = 2–4 for MSM.

How a limited selection of demographic traits related to partner counts, by partnership type, is described in univariate analysis for interested readers in S10 Table in S1 Appendix. The full survey dataset is available for persons interested in focusing on other relationships with demographic traits (not just age), different sub-samples or research questions.

## Results

Table 1 shows selected demographic traits of respondents, divided by recruitment sample. Note that recruitment sample totals (final row) are the maximum number of possible respondents, not actual totals that gave data for each question. Grindr respondents were younger on average than all others. Living in a high deprivation area did not vary markedly between recruitment samples (22%-34%). The Grindr and Meta respondents were much more likely to have a university degree, while there were more skilled, semi-skilled, unskilled workers, retired or keeping house persons in the general population recruitment sample. The social media samples were much less likely to have a dependent child or find it very/extremely difficult to pay bills, while being much more likely to be London residents. A small majority (52%) of all respondents had a regular partner, although this was lowest in the Grindr sample (32%) and highest in the general population recruitment sample (58%). Identifying as a non-white ethnicity did not vary greatly by recruitment sample (6.1–9.7%).

We identified 2132 MSM, 1201 MSW and 1308 WSM. There was overlap in MSW and MSM, n = 340, 28% of MSW were also MSM. This proportion (28%) is higher than the estimates of how many MSM are also MSW as reported in other recent surveys (13–19%; [21–24]). Some of those sources relied on only behaviour or only sexual orientation rather than orientation *and* reported behaviour, or asked about different time periods (not last 3 months), making it hard to know if our results or their statistics are more indicative of the true overlap in general population.

The proportions of respondents in specific age groups with zero, one, or 2+ partners in preceding three weeks are shown in Fig 1A–1C. MSM are divided in these figures by sample origin, Savanta only (presupposed to be more representative of general population MSM) and other sample (from social media advertising, presumed to be highly sexually active). Denominators are only for respondents who gave an answer (missing data respondents omitted). One similarity between groups is that the proportion with zero partners increased with age, but much faster/higher for WSM than the male groups. WSM have a near linear increase of the percentage with zero partners (Fig 1A), corresponding to a near linear decline in percentage with exactly one partner (Fig 1B) from about age 40 onwards. In contrast, the proportion of MSW and MSM with exactly one partner was fairly consistent at all ages, about 50% for MSW

**Table 1. Demographics of survey respondents by recruitment sample group.**

| Trait \ Sample | Savanta GP n = 3050 | Savanta B n = 247 | Meta n = 1036 | Grindr n = 831 | Total n = 5164 |
|---|---|---|---|---|---|
| **Male** | 42% | 100% | 100% | 100% | 66% |
| **Age, median: IQR** <br> **Full range** | 49: 34–65 <br> 18–98 | 48: 33–61 <br> 18–77 | 48: 39–56 <br> 18–79 | 44: 35–53 <br> 18–82 | 48: 35–60 <br> 18–98 |
| **Most deprived quintile** | 34% | 34% | 22% | 28% | 31% |
| **University degree** | 32% | 36% | 76% | 64% | 46% |
| **Has dependent child** | 32% | 15% | 2% | 4% | 21% |
| **Has regular partner** | 58% | 43% | 53% | 32% | 52% |
| **Very or extremely difficult to pay bills** | 27% | 25% | 6.6% | 12.9% | 20.5% |
| **Identify as non-white** | 9.0% | 6.1% | 6.1% | 9.7% | 8.4% |
| *Region* | | | | | |
| **East Midlands** | 7.6% | 6.5% | 3.5% | 3.8% | 6.1% |
| **East of England** | 10.1% | 8.9% | 5.5% | 6.0% | 8.5% |
| **London** | 10.1% | 18.6% | 36.6% | 23.5% | 18.0% |
| **North East** | 4.4% | 4.0% | 2.0% | 2.0% | 3.5% |
| **North West** | 12.0% | 13.0% | 8.3% | 7.9% | 10.6% |
| **Northern Ireland** | 1.6% | 0.8% | 0.7% | 1.8% | 1.4% |
| **Scotland** | 8.0% | 8.1% | 4.7% | 6.7% | 7.1% |
| **South East** | 13.9% | 12.6% | 14.6% | 12.5% | 13.8% |
| **South West** | 8.5% | 6.9% | 6.8% | 6.4% | 7.7% |
| **Wales** | 5.3% | 4.9% | 2.7% | 2.9% | 4.4% |
| **West Midlands** | 9.5% | 9.3% | 4.7% | 4.9% | 7.8% |
| **Yorkshire & Humber** | 9.0% | 6.5% | 4.6% | 4.3% | 7.3% |
| *Missing* | 0.0% | 0.0% | 5.2% | 17.1% | 3.8% |
| *Occupational group* | | | | | |
| **HMAP** | 6.3% | 11% | 24% | 17% | 12% |
| **Int** | 20% | 23% | 37% | 37% | 26% |
| **Sup** | 21% | 21% | 19% | 21% | 20% |
| **Other** | 24% | 14% | 5% | 11% | 17% |
| **Unemp** | 5.8% | 9.8% | 2.5% | 3.3% | 4.9% |
| **Student** | 1.7% | 2.0% | 3.1% | 3.6% | 2.3% |
| **Retired/KH** | 22% | 18% | 10% | 7% | 17% |

*Notes*: Age in years. *Occupation groups*: HMAP: Higher managerial or administrative professional, Int: Intermediate managerial or administrative professional; Sup: Supervisor, administrative or uniformed professional; Other: Skilled, semi-skilled manual or unskilled workers; Unemp: Unemployed; Student: University or college; Retired/KH: Retired or keeping house. (Savanta) GP: General population; B: Boost sampling of UKMSM.

and Savanta sample MSM, but was lower (19–33%) for the social media samples (Fig 1B). Partner concurrency ($\geq$ 2 partners in last 3 weeks) was low for MSW and WSM, maximum 14% at age 18–24 (MSW). Concurrency for MSW and WSM declined with age and was especially low (0–2%) after age 60. Partner concurrency varied by which sample the MSM were in. Just 15% (25/161) Savanta sample MSM had 2 or more recent partners, peaking at 34% partner concurrency for age 18–24. In contrast, partner concurrency for social-media sampled MSM was steadily 45–51% for those aged 18–64, after which reported concurrency percentages declined to 31%.

Table 2 shows the mean partner count for respondents by partnership type, with distinctions by recruitment origin for MSM. Average partner counts were highest for under 30s

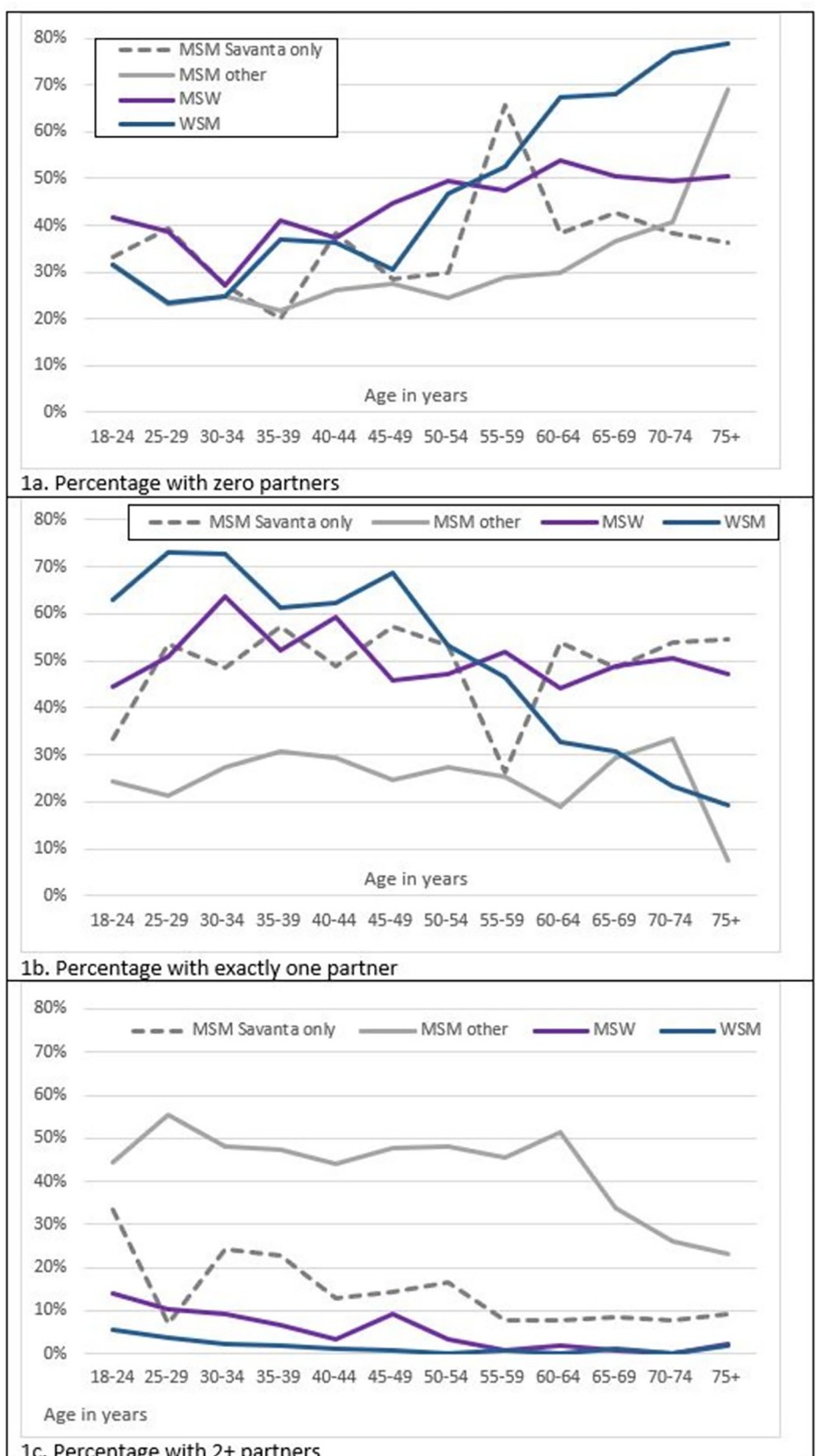

**Fig 1. Percentage of respondents with 0, 1 or 2+ partners, by partnership type.**

**Table 2. Average partner count by partnership type, sub samples by recruitment origin for MSM.**

| Age group (years) | n = 369 MSM Savanta | n = 980 MSM Meta | n = 783 MSM Grindr | n = 1201 MSW | n = 1285 WSM |
|---|---|---|---|---|---|
| 18–24 | 1.61 | 1.61 | 2.74 | **1.23** | 0.86 |
| 25–29 | 1.21 | 2.42 | 2.58 | 1.19 | **1.01** |
| 30–34 | 1.24 | 1.93 | 2.57 | 0.90 | 0.78 |
| 35–39 | **1.80** | **3.13** | 2.31 | 0.87 | 0.65 |
| 40–44 | 0.74 | 3.02 | 3.19 | 0.75 | 0.65 |
| 45–49 | 1.07 | 2.94 | 2.84 | 0.77 | 0.72 |
| 50–54 | 1.23 | 2.51 | 3.36 | 0.61 | 0.53 |
| 55–59 | 0.61 | 2.34 | 2.57 | 0.54 | 0.48 |
| 60–64 | 1.00 | 2.06 | **3.42** | 0.49 | 0.33 |
| 65–69 | 0.83 | 1.83 | 3.37 | 0.50 | 0.44 |
| 70–74 | 0.69 | 1.32 | 1.80 | 0.50 | 0.23 |
| 75+ | 0.82 | 1.43 | 1.67 | 0.52 | 0.25 |

**Note**: Highest values in bold green font.

among MSW and WSM, followed by apparent linear decline with age. In contrast, average partner counts were highest for persons aged 35–54 among MSM (aged 35–64 for MSM recruited via Grindr). Among MSM recruited via social media, persons aged 65–69 had higher average partner counts than persons aged 18–24 years.

Tables 3–5 show the Bayesian Information Criterion (BIC) scores for the respective partnership types and partner count, with different expressions of age as predictor in both univariate and models adjusted for recruitment sample, occupation group, region, ethnic group, deprivation quintile, ability to pay bills, education level, if they had a dependent child, and existence of regular partner or not. Weibull models have the lowest BIC consistently. For all

**Table 3. Bayesian Information Criterion (BIC) values for adjusted models and stated expressions of age variable in partner count prediction models: MSM.**

| Model form | Count of respondents | p-value for age expression | Age expression | BIC |
|---|---|---|---|---|
| *Univariate models* | | | | |
| Negative binomial | 2132 | 0.223 | Linear | 8752 |
| Negative binomial | 2132 | < **0.001** | Quadratic (age × age) | 8732 |
| Negative binomial | 2132 | 0.543 | Reciprocal × population mean | 8753 |
| Weibull | 1518 | 0.496 | Linear | 4349 |
| Weibull | 1518 | < **0.001** | Quadratic (age × age) | 4341 |
| Weibull | 1518 | 0.060 | Reciprocal × population mean | 4346 |
| *Adjusted models* | | | | |
| Negative binomial | 1928 | 0.832 | Linear | 8102 |
| Negative binomial | 1928 | < **0.001** | Quadratic (age × age) | 8096 |
| Negative binomial | 1928 | 0.151 | Reciprocal × population mean | 8100 |
| Weibull | 1379 | 0.192 | Linear | 4105 |
| Weibull | 1379 | < **0.001** | Quadratic (age × age) | **4096** |
| Weibull | 1379 | **0.014** | Reciprocal × population mean | 4100 |

**Notes for Tables 3–5:** Adjusted models were all adjusted by the same demographic variables, as reported in main manuscript. Models are for all age respondents in this partnership type. Count of respondents is dependent on complete information for all correlates and whether zero-partner count observations were excluded. The lowest BIC is highlighted in **bold green font.**

**Table 4. Bayesian Information Criterion (BIC) values for adjusted models and stated expressions of age variable in partner count prediction models: MSW.**

| Model form | Count of respondents | p-value for age expression | Age expression | BIC |
|---|---|---|---|---|
| *Univariate models* | | | | |
| Negative binomial | 1201 | < **0.001** | Linear | 2696 |
| Negative binomial | 1201 | **0.005** | Quadratic (age × age) | 2701 |
| Negative binomial | 1201 | < **0.001** | Reciprocal × population mean | 2700 |
| Weibull | 660 | < **0.001** | Linear | 1315 |
| Weibull | 660 | < **0.001** | Quadratic (age × age) | 1304 |
| Weibull | 660 | < **0.001** | Reciprocal × population mean | 1301 |
| *Adjusted models* | | | | |
| Negative binomial | 1161 | < **0.001** | Linear | 2712 |
| Negative binomial | 1161 | < **0.001** | Quadratic (age × age) | 2710 |
| Negative binomial | 1161 | < **0.001** | Reciprocal × population mean | 2704 |
| Weibull | 642 | < **0.001** | Linear | -687 |
| Weibull | 642 | 0.081 | Quadratic (age × age) | -686 |
| Weibull | 642 | < **0.001** | Reciprocal × population mean | **-688** |

**Notes:** See notes for Table 3.

MSM, only the quadratic expression of age had a significant relationship with partner count (Table 3) in all model forms. The strong quadratic relationship indicates higher partner counts for MSM in the mid-life period, as evidenced in Table 2.

Quadratic age also had a somewhat more consistent relationship (than linear or population mean adjusted) with partner count for WSM. For MSW, all trialled expressions of age had significant relationship with expected partner counts, with reciprocal × population mean adjusted having the lowest BIC for each model form tested. This result may reflect over-sampling of older age men in the recruitment strategy. The Weibull model outperformed the negative binomial, indicating a 'heavy tail' in partner counts for all partnership types.

**Table 5. Bayesian Information Criterion (BIC) values for adjusted models and stated expressions of age variable in partner count prediction models: WSM.**

| Model form | Count of respondents | p-value for age expression | Age expression | BIC |
|---|---|---|---|---|
| *Univariate models* | | | | |
| Negative binomial | 1308 | < **0.001** | Linear | 2519 |
| Negative binomial | 1308 | < **0.001** | Quadratic (age × age) | 2521 |
| Negative binomial | 1308 | < **0.001** | Reciprocal × population mean | 2544 |
| Weibull | 691 | 0.316 | Linear | 1238 |
| Weibull | 691 | < **0.001** | Quadratic (age × age) | 1222 |
| Weibull | 691 | **0.020** | Reciprocal × population mean | 1233 |
| *Adjusted models* | | | | |
| Negative binomial | 1306 | < **0.001** | Linear | 2563 |
| Negative binomial | 1306 | < **0.001** | Quadratic (age × age) | 2569 |
| Negative binomial | 1306 | < **0.001** | Reciprocal × population mean | 2567 |
| Weibull | 691 | 0.562 | Linear | 921 |
| Weibull | 691 | **0.007** | Quadratic (age × age) | **919** |
| Weibull | 691 | 0.431 | Reciprocal × population mean | 920 |

**Notes:** See notes for Table 3.

## Discussion

With increased age, a constant decline in partner count was evident in heterosexual partnership types, but a quadratic (peaking in middle age) relationship describes MSM partner count patterns better. These survey data also suggest that MSM were much more likely to have higher concurrency at all ages, and to be sexually active at age 65+ than WSM or MSW. Peak likelihood of concurrency tended to be about age 35–54 for MSM when taking into account sampling strategies and different models. In addition to improving disease model accuracy, considering more realistic partner count variations arising from age distributions in real populations could be useful when trying to design awareness and intervention strategies.

The survey data we used provide a snapshot of sexual behaviour in the UK in September-October 2022. We encourage others to access the dataset and use it to explore other research questions, but with awareness of many caveats. Among MSM at least, it is possible that partner counts were lower than usual. The survey was administered when risks of catching mpox had been well-publicised in UK and public health campaigns targeting MSM were encouraging them to reduce some forms of sexual activity. 60% of MSM, 30% of MSW and 22% of WSM answering our survey said they had heard "a lot" about mpox [7]. 49% of American MSM answering an Internet survey in August 2022 [25] said they were limiting their partner count to reduce their chances of catching mpox; it seems likely similar partner reduction was true of many of the MSM in our UK survey. Evidence about the dynamics of the mpox epidemic in 2022 in different countries is still emerging, but early opinion was that behaviour change may have been an important reason for decline of the outbreak [26], and probably more important than delivery of the smallpox vaccine (which can prevent disease development and transmission). It is likely that the vaccine did not strongly reduce the epidemic because relatively few vaccine doses were delivered before new mpox diagnoses started to decline in Britain [27]. Behaviour change also would explain observed reduction during the mpox outbreak in new diagnoses of other sexually transmitted infections (STI) in the MSM population [15]. How much activity needs to be reduced to lower numbers of new cases is still to be explored; it may be that only relatively small reductions in partner count, concurrency or sex acts were enough to bring the epidemic under control. Although median partner counts for MSM were higher than for MSW and WSM, many MSM were not recently sexually active: a substantial minority of MSM respondents (26%) reported no partners in the preceding 3 weeks.

The age distributions of our respondents varied by recruitment sample and by partnership group (MSM, MSW or WSM). As described previously, the Savanta sampling strategy was balanced for deprivation, age groups and geographic regions. Hence, the Savanta sampling recruited relatively more persons aged 18 to 24 or 65+ than responded via the social media adverts. The reported partner counts for respondents aged 65+ was significantly lower than for persons aged < 65 years, but the partner counts did not reduce to zero at age 65+. Often in MSM disease transmission modelling, age is treated as a key determinant of sex act frequency. The focus in STI models has been on adults below about age 45 years, so much so that individuals near age 65+ may be retired out of the presumed sexually active population in models of MSM behaviour [3,4,28,29]. Our data and analysis support more sophisticated modelling of partner activity for persons age 65+. We report our complete information across the spectrum of ages surveyed, broken down by MSM, MSW and WSM subgroups, to help modellers make informed decisions based on empirical data about number of partners and total sexual activity related to their age, adjusting for other demographic traits. It is a useful finding that the data indicate a very different relationship between age and partner count for MSM compared to MSW or WSM. It was also useful to note that the median partner count across all groups was one, with a large number of respondents reporting no recent partners.

## Limitations

How representative any survey data are for each of MSM, MSW and WSM communities in the general population needs to be considered with regard to the specific demographic distributions obtained and potential sampling biases. Our data were reliant on self-report. This may have led to inflated responses in a very small number of participants (due to mischievousness or error). However, it seems more likely to have led to under reporting, although the anonymity of an online survey should have mitigated this to some extent. Rounding was evident in the partner count responses (a tendency to answer 10, 15, 20, 30 or 40 etc. instead of stating numbers in between). Data cleaning suggested some responses were mistakes; other data entry mistakes probably existed and were not detected. The sample of WSM only found 20 individuals with $> 1$ partner, which weakens confidence in the suggested correlations in the WSM models. We did not consider many possible interesting sub-groups such as men who only have sex with women, men who have sex with both men and women, and so on. With awareness of caveats we have listed, the survey data that we put in public domain enables others to explore those subgroups at length for other research questions. Probably because of our sampling strategy, most of the MSM in our study who were also MSW had many more male partners or no female partners at all, which means that an analysis of a total partner count for MSW&M using our data would need to be undertaken very carefully. There were demographic differences between the partnership groups that cannot be obviously explained by partnership group membership and that suggest potentially important differences between recruitment samples: for instance, significantly more MSW than MSM were retired (23.5% vs. 9.8%), which is a statistically significant difference: $\chi^2(1) = 81.7$, $p < 0.001$. We do not have data to confirm how representative of the full UK MSM, MSW or WSM population the recruitment samples were with respect to the model outcomes (reported partner counts). We have not tested in transmission models or in an exemplar population if varying expected partner count by age makes significant differences to key outcomes such as total persons made ill or relative effectiveness of possible disease prevention strategies.

A strength of our analysis is that it used an especially large sample, with over 1000 MSW and WSM respondents and over 2000 MSM. For comparison, a recent mpox transmission model was generated using data about partner counts from just a few hundred UK MSM respondents [2]. Our data can be divided into useful sub-samples, some of which (such as the Savanta groups) were purposefully balanced with respect to deprivation, geography and age distribution. The Grindr recruitment sample indicates behaviour patterns by a likely high-risk group at a key moment in the UK mpox outbreak. The data empirically confirm that, contrary to some modelling assumptions, many persons age 65+, especially MSM, have not retired from sex.

## Supporting information

**S1 Appendix. Distribution of partner counts.**
(DOCX)

## Acknowledgments

Elizabeth Fearon at the UCL Institute for Global Health suggested collection of data about partner counts, how to phrase the questions and time boundaries to use. Thanks to the Terrence Higgins Trust for support with participant recruitment, and Savanta for hosting the survey. We are grateful to study participants for their honesty and time, and to the Internet for having so many examples of how to do useful things in Stata.

## Author Contributions

**Conceptualization:** Julii Brainard.

**Data curation:** Julii Brainard, Louise E. Smith.

**Formal analysis:** Julii Brainard.

**Funding acquisition:** G. James Rubin.

**Investigation:** Louise E. Smith.

**Methodology:** Julii Brainard.

**Project administration:** Louise E. Smith, G. James Rubin.

**Software:** Julii Brainard.

**Supervision:** G. James Rubin.

**Validation:** Julii Brainard, Henry W. W. Potts.

**Visualization:** Julii Brainard.

**Writing – original draft:** Julii Brainard.

**Writing – review & editing:** Julii Brainard, Louise E. Smith, Henry W. W. Potts, G. James Rubin.

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
