## [Decision Letter · Decision Letter 0]

11 Jul 2023

PONE-D-23-04595The relationship between age and sex partner counts during the mpox outbreak in the UK, 2022PLOS ONE

Dear Dr. Smith,

Thank you for submitting your manuscript to PLOS ONE. After careful consideration, we feel that it has merit but does not fully meet PLOS ONE’s publication criteria as it currently stands. Therefore, we invite you to submit a revised version of the manuscript that addresses the points raised during the review process.

We look forward to receiving your revised manuscript.

Kind regards,

Soham Bandyopadhyay

Academic Editor

PLOS ONE

Journal Requirements:

"The study was funded by the National Institute for Health and Care Research (NIHR). This work was funded by the National Institute for Health and Care Research Health Protection Research Unit (NIHR HPRU) in Emergency Preparedness and Response, a partnership between the UK Health Security Agency (UKHSA), King’s College London (KCL) and the University of East Anglia (UEA).  The views expressed are those of the authors and not necessarily those of the NIHR, UKHSA, Department of Health and Social Care, UEA, KCL, or University College London (UCL). For the purpose of open access, the author has applied a Creative Commons Attribution (CC BY) licence to any Author Accepted Manuscript version arising."        

"HWWP receives consultancy fees to his employer from Ipsos MORI and has a PhD student who works at and has fees paid by AstraZeneca. All other authors declare that we have no conflict of interest."

5. We noted in your submission details that a portion of your manuscript may have been presented or published elsewhere. [Data from the same dataset have been submitted for publication elsewhere. Those analyses investigate mpox beliefs, knowledge and intended behaviours. A pre-print of that article has been made available at https://doi.org/10.1101/2022.12.07.22283201.] Please clarify whether this [conference proceeding or publication] was peer-reviewed and formally published. If this work was previously peer-reviewed and published, in the cover letter please provide the reason that this work does not constitute dual publication and should be included in the current manuscript.

Reviewers' comments:

Reviewer's Responses to Questions

**Comments to the Author**

1. Is the manuscript technically sound, and do the data support the conclusions?

Reviewer #1: Yes

Reviewer #2: Partly

2. Has the statistical analysis been performed appropriately and rigorously? 

Reviewer #1: Yes

Reviewer #2: No

3. Have the authors made all data underlying the findings in their manuscript fully available?

Reviewer #1: Yes

Reviewer #2: No

4. Is the manuscript presented in an intelligible fashion and written in standard English?

Reviewer #1: Yes

Reviewer #2: No

5. Review Comments to the Author

Reviewer #1: Overall the study is well written and data are presented in a good way. Bellow are some points that the author should follow and correct the manuscript accordingly.

1. The introduction needs to have more literature on this subject.

2. In the introduction, the authors should clearly address the aim and objectives of the study.

3. From lines 45-68 of the introduction, the authors are addressing the methodological part of their work and also list some results. These contents are inappropriately placed in the introduction part and should be removed.

4. In line 45, the authors are explaining the survey they have conducted and are citing an article, what is that reference?

5. How the authors can link their study outcomes to the mpox epidemic?

6. In conclusion, how these results could be used for awareness to control or slow the epidemic?

Reviewer #2: In this manuscript, Dr. Smith and colleagues explored the relationship between age and the number of sexual partners for MSM, MSW and WSM, constructing a larger and more recent database than existing studies. They found all the sexual partnership distributions are skewed but they exhibited different patterns in the relationship between partner counts and age: a linear declining relationship for WSM and a quadratic relationship for MSW and MSM. Overall, the study is well described and I think this study has a potential to be interesting and valuable. However, at present there are several critical issues and minor points that need to be addressed.

[Major comments]

1. The authors undertook a survey in 5 September-6 October, highlighting their survey is more recent than other survey in the UK (e.g., Natsal-3) in the introduction section. However, Natsal-4 (https://www.natsal.ac.uk/natsal-survey/natsal-4) was conducted in September 2022 and is supposed to be more recent than this study. While I understand that it was not possible to include it in this study because this new dataset has not been made public, I encourage the authors not to focus too much on the recency of their data when claiming the merit of the study. (Because, if that is the primary merit of the study, it means that the merit would be lost when Natsal-4 is published; which I believe is not the intention of the authors) If possible, it would also be helpful to briefly mention/discuss Natsal-4 as this will give the readers useful reference and context.

2. Because it is an explanatory data analysis (as stated by the authors), I would recommend that the model selection should be based on BIC, not AIC. AIC is constructed by approximating generalization loss (or predictive performance; i.e. the divergence between the true distribution and predictive distribution) while BIC is based on model evidence, which indicates that BIC is more useful in selecting a correct model while the AIC is more appropriate in finding the best model for predicting future observations (https://doi.org/10.1016/B978-0-444-51862-0.50018-6).

3. I found that the authors’ definitions of MSM/MSW/WSM do not exclude sexually inactive people, who are now also counted as those having zero partner. This means “zero-partner individuals” comprise (i) sexually active persons but with no sexual contact over the study period, and (ii) sexually inactive persons, which may be problematic in the estimation. If the authors can distinguish those two based on the original data, they should simply exclude (ii), sexually inactive persons, from their analyses. If this is not possible, I think there are some options to address this issue. One is to left-truncate the distributions at partner count=1 to ignore zero counts. Alternatively, they can also use zero-inflated distribution, which enable the authors to allow for the sexually inactive persons statistically.

4. For the sentence at P5L128-130 “Among respondents who gave the correct answer to the attention check question, manual inspection of free-text responses to four questions did not suggest that any of these were generated by artificial intelligence algorithms.”, please elaborate on this. How did the authors distinguish human and AI? What kind of AI did they consider?

5. The authors utilized a negative-binomial distribution across all the groups (MSM/MSW/WSM) but using the same ad-hoc distribution for all cases might not be fully justified as each group may have a unique sexual partnership pattern. Besides, sexual distributions have been reported to typically have heavy-tailed or power-law tailed property in the large body of literatures (e.g., Schneeberger et al. https://doi.org/10.1097/00007435-200406000-00012, Endo et al. https://doi.org/10.1126/science.add4507, Ito et al. https://doi.org/10.1371/journal.pone.0221520). These studies suggest that a negative binomial may not fully capture the (extreme) level of heterogeneity in sexual partners. I’d like the authors to reanalyse all the groups, using heavy-tailed distributions such as Pareto or Weibull distributions, visually inspect the fit, and/or select a best-fit model for each group by BIC.

6. What type of distributions did the author employ in the regression models with which they tried such as hurdle regression, zero-inflated models, etc (described around P6L186)? Did the authors only use a negative binomial distribution to the other models? On that note, I do not generally agree with the “data not shown” practice the authors use here, because the statements cannot be interpreted without them along with full description of the methods. Please include in the Supplementary file.

7. Regarding the sentence, “The statistical model form reported here (negative binomial) was preferred because the outputs could be easily reported and replicated, supporting development of realistic mpox and sexually transmitted infection models.” at P7L188-191, I think this does not support their model choice at all since there is no statistical justification matched with data. I’m also not sure about what the authors imply by “because the outputs could be easily reported and replicated, supporting development of realistic mpox and sexually transmitted infection models”. As mentioned above, my understanding is that sexual partner data have been rarely characterized by negative binomial distributions.

8. Although the authors repeatedly highlighted their findings are useful in modelling sexually-associated transmissions, there is no description as to how it would be useful. I would be interested in how the authors’ findings could be plugged into disease models. I am not sure if data-driven network modelling for STIs allowing for age-dependent heterogeneity in sexual contact networks is possible as the authors suggest when there is little empirical data on age assortativity in sexual partner formation.

9. Related to #6; Although the authors said they decided not to present other model approaches (P6L186), I encourage them to show the comparison of all of the fitting including model selection to justify their choice of the final model.

10. It would be very informative to show log-log plots for the sexual partnership distributions. This would make readers easily understand how each distribution is skewed.

11. According to P8L211-212, 28% of MSW are also MSM in the data, which is fairly high compared to the actual proportion of the overlap. This is not surprising because MSM were oversampled due to the nature of the surveys except for Savanta GP, but for this reason I am not sure how we should interpret this 28% figure. I suggest the authors consider alternative ways to characterize the overlap between MSW and MSM.

12. The authors calculated median of partner counts in each cohort but, as I pointed out in the comment #3, the authors may wish to address the issue of people with zero partner counts when analyzing them (because the median would be sensitive to handling of these people). Probably excluding people with zero partner count would be enough for the purpose of quick comparison.

13. In the discussion section (P9L247-P10L265), the authors discussed that the drastic decline in mpox cases would be primarily attributed to behavioural changes citing some papers. However, there is another key factor that would play significant roles in shaping mpox epidemic sizes: that is, depletion of susceptibles effect over heavy-tailed sexual contact networks (Murayama et al. https://doi.org/10.1093/infdis/jiad254; Xiridou et al. https://doi.org/10.1101/2023.01.31.23285294). These studies suggest that accumulation of infection-derived immunity in heavy-tailed sexual networks can dramatically lower the herd immunity threshold and final sizes, and the observed decline in cases may not be primarily attributable to behavioural changes or interventions. I’d like more in-depth discussion in the manuscript that touches on the depletion of susceptibles effect. As a side note, ref 15 also discussed this effect as “the biggest factor”. So, it’d be inappropriate to say “early opinion was that decline in the outbreak was more due to behaviour change (15)”, citing ref 15.

14. I do not think discussing the impact of behavioural changes citing ref 14 without noting its limitations is a good idea. Ref 14 did not well inform the impact of behavioural changes on the transmission dynamics as they did not use quantitative measures, alongside that it is unclear from the study whether the results are representative of people who have many partners and are thus playing roles in mpox transmission.

15. Please discuss the strength of the study more convincingly than the authors currently state at P11L304-306. It is indeed a good point that the existing mpox modeling study (ref 2) relied on relatively limited MSM samples, but the authors did not clarify what changes their new data could bring to such studies. For example, the authors’ data indeed has 4x sample size than the data used in ref 2, but is it enough to significantly improve the results of that study (yes confidence intervals may be slightly narrower but is it just that?)? I would be convinced if the authors’ data can provide detailed information at the tail part of the sexual partner distributions that had not been available with small sample data. Other aspects of the authors’ data, including sub-samples and the presence of active 65+ individuals, would also be potentially useful in modeling but the authors did not discuss how. Meanwhile, it should also be noted that bias in sampling may undermine the strength of a large sample size. Savanta may be representative as they used quota sampling method, but other cohorts, in particular Grindr, are not. Then the sample size in Savanta becomes 369 (=161+208) and it is almost the same order (a few hundreds the authors said to ref 2 at P11L306). Rather than putting too much emphasis on the sample size, I think it’d be better to put more focus on their detailed information about each data (represented in Table S1-S9), which would be beneficial for those who attempt to construct an age-specific network model.

[Minor comments]

1. The authors often described MSM/MSW/WSM as MSMs/MSWs/WSMs, but “s” should be deleted.

2. I recommend the authors make the overall writing more formal, objective, and quantitative throughout. There are a number of instances where the text is rather casual, empty, or vague . I present some examples below:

“We wanted to-” at P2L56,

“Grindr respondents were somewhat younger-“ at P7L195,

“it is quite possible-” at P9L248,

etc.

3. Typo: “low frequency sexual contact” at P4L122 -> “low frequency of sexual contact”

4. Ref 14 is a summary article based on the original research paper. Please cite this original paper instead (http://dx.doi.org/10.15585/mmwr.mm7135e1).

5. In light of the reproducibility of this study and also for the study to make an impact on future research, I hugely encourage the authors to share the data and stata code they made on a platform such as GitHub repository. It is stated that the data will be shared along with another paper but this only makes sense if that paper is published before this paper is published (and I am not fully sure of the point of safeguarding the data when the preprint is already public…?)

6. Please do not use abbreviations in Figures (e.g. ptnrs to partners).

6. PLOS authors have the option to publish the peer review history of their article (what does this mean?). If published, this will include your full peer review and any attached files.

Reviewer #1: **Yes: **Jivan Qasim Ahmed

Reviewer #2: No

---

## [Author Response · Author response to Decision Letter 0]

10 Aug 2023

Editor comments

This Rebuttal/response letter is supplied.

We have uploaded 2 manuscripts as described

We found the templates and revised accordingly

A statement was added to Methods section that stated that consent was digital and written, and names the approval body. The survey was administered only to adults so minors not involved.

"The study was funded by …." 

We added this statement to our cover letter

4. Thank you for stating the following in the Competing Interests section ….

Please confirm that this does not alter your adherence to all PLOS ONE policies on sharing data and materials, by including the following statement: "This does not alter our adherence to PLOS ONE policies on sharing data and materials.” 

We added this phrase to our cover letter

Note that it is not acceptable for the authors to be the sole named individuals responsible for ensuring data access.

We will update your Data Availability statement to reflect the information you provide in your cover letter. The data are available in a public repository, not from a sole author.

With regard to data availability, we now make this statement:

See file = partner count.dta at https://osf.io/p5f6y/. 

We added the necessary phrase to our cover letter

5. We noted in your submission details that a portion of your manuscript may have been presented or published elsewhere. …. Please clarify whether this [conference proceeding or publication] was peer-reviewed and formally published. If this work was previously peer-reviewed and published, in the cover letter please provide the reason that this work does not constitute dual publication and should be included in the current manuscript. We also added the below information to the revised cover letter. 

The introduction now makes clearer that this is a secondary analysis of data collected in a different study (cited); the data collection was originally part of a health messaging experiment (RCT) and cross sectional knowledge survey.

The original study is available in preprint & is undergoing peer-review (we will cite the peer-review version if available in time).

The reason that this submission is not dual publication is that although the same data are available in the primary study, that study does not undertake a remotely similar analysis of partner counts by partnership type.

The minimal underlying dataset is now on a public repository, indicated with short URL name in manuscript 

The data are available in a declared public repository

We have no information that indicates any of our cited articles were retracted, therefore no changes were made

Reviewers' comments:

Reviewer #1: Overall the study is well written and data are presented in a good way.

Thank you for this positive comment.

1. The introduction needs to have more literature on this subject.

We added 7 new citations to the Introduction.

2. In the introduction, the authors should clearly address the aim and objectives of the study.

Explicit aim and objectives statements are now in the introduction. This change helps to address many points, including point 3 (next)

3. From lines 45-68 of the introduction, the authors are addressing the methodological part of their work and also list some results. These contents are inappropriately placed in the introduction part and should be removed.

We understand why reviewer said this, and rewrote narrative, especially Introduction, to explicitly explain that our study is a secondary analysis of data collected for another study. 

Part of this narrative is about typical assumptions in previous models, and also about the published ages of mpox cases in UK (not our data but rather contextual information to help justify the study). So the information is not about our own data nor states our own results; rather, the information sets context for why exploring relationship (better modelling of) between age and partner count for each partnership type could be good thing

4. In line 45, the authors are explaining the survey they have conducted and are citing an article, what is that reference?

Apologies, this reference (to original study that collected data used in this secondary analysis) was missing/misformed, is now correctly cited.

5. How the authors can link their study outcomes to the mpox epidemic?

The Introduction lists some age statistics about the 2022 mpox cases in the UK, which data support our hypothesis that age may have a complicated relationship with partner counts.

We note in the Limitations that we haven’t applied the study results in our own models (we would like to do that in a follow up study).

6. In conclusion, how these results could be used for awareness to control or slow the epidemic? 

We added several sentences that may be relevant, including this one in Discussion:

“In addition to improve disease model accuracy, considering partner count variations arising from age distributions in real populations could be useful when trying to design awareness and intervention strategies.”

Reviewer #2 

Overall, the study is well described and I think this study has a potential to be interesting and valuable.

Thank you for this positive comment and for many specific suggestions after thorough reading

Major comments 

1. The authors undertook a survey in 5 September-6 October, highlighting their survey is more recent than other survey in the UK (e.g., Natsal-3) in the introduction section. However, Natsal-4 was conducted in September 2022 and is supposed to be more recent than this study. While I understand that it was not possible to include it in this study because this new dataset has not been made public, I encourage the authors not to focus too much on the recency of their data when claiming the merit of the study. 

Thank you for this information. The link says that data collection for Natsal-4 will continue into 2023. We have revised our text to not say that our study is more recent than Natsal-4.

Please see revised text 1st paragraph of Introduction

If possible, it would also be helpful to briefly mention/discuss Natsal-4 as this will give the readers useful reference and context.

We have now mentioned Natsal-4 (with URL)

Partly Because its results aren’t published and we didn’t know if the survey instrument was same as Natsal-3, we agree with reviewer that we shouldn’t compare our own survey too much with Natsal, we don’t know enough about what we would be comparing to.

2. Because it is an explanatory data analysis (as stated by the authors), I would recommend that the model selection should be based on BIC, not AIC. AIC is constructed by approximating generalization loss (or predictive performance; i.e. the divergence between the true distribution and predictive distribution) while BIC is based on model evidence, which indicates that BIC is more useful in selecting a correct model while the AIC is more appropriate in finding the best model for predicting future observations (https://doi.org/10.1016/B978-0-444-51862-0.50018-6).

We were happy to use BIC instead.

3.1 I found that the authors’ definitions of MSM/MSW/WSM do not exclude sexually inactive people, who are now also counted as those having zero partner. This means “zero-partner individuals” comprise (i) sexually active persons but with no sexual contact over the study period, and (ii) sexually inactive persons, which may be problematic in the estimation. If the authors can distinguish those two based on the original data, they should simply exclude (ii), sexually inactive persons, from their analyses.

This is a reason we previously tried hurdle models, with zero-truncated negative binomial regression for the 2nd stage: persons with partner count > 1. Technically to exclude the zero-count people would be to make assumptions about them that we couldn’t evidence robustly (we didn’t ask if they are celibate by choice or circumstances), and there was no theoretical or stats basis to prefer the hurdle models otherwise (not a much better AIC/BIC).

We agree on the value of trying to recognise this problem, as well as how excluding the zero-partner count persons could be handled in sensitivity analysis about the apparent relationship between age and partner counts. Therefore the revised article reports on two model approaches : negative binomial regression on full data by partnership type, as well as applying Weibull distribution with several expressions of the age variable, form of which is our primary focus

3.2 If this is not possible, I think there are some options to address this issue. One is to left-truncate the distributions at partner count=1 to ignore zero counts. Alternatively, they can also use zero-inflated distribution, which enable the authors to allow for the sexually inactive persons statistically.

We report on two alternative model forms as suggested, negative binomial regression on full data by partnership type, with several expressions of the age variable which is our primary focus, as well as applying Weibull distribution to non-zero respondents. These models are applied in order to pursue our clarified aim which is to explore the relationship between Age and reported partner counts. We don’t consider a large variety of other models because we didn’t set out to find the best possible model fit. We do provide the underlying dataset to enable other interested researchers to undertake that analysis if they like.

4. For the sentence at P5L128-130 “Among respondents who gave the correct answer to the attention check question, manual inspection of free-text responses to four questions did not suggest that any of these were generated by artificial intelligence algorithms.”, please elaborate on this. How did the authors distinguish human and AI? What kind of AI did they consider?

We Rewrote the sentence as below; it was a subjective judgement based on our previous experiences of learning how to identify ‘bot’ responses in other surveys. To be honest, we don’t like to say too much publicly about how to identify bot answers because that would help the bot writers evade detection.

“Among respondents who gave the correct answer to the attention check question, manual inspection of free-text responses to four questions did not suggest that any of these were nonsensical, illogical in the UK context or formulaic sounding which could happen because they were generated in bad faith or by artificial intelligence algorithms.”

5. The authors utilized a negative-binomial distribution across all the groups (MSM/MSW/WSM) but using the same ad-hoc distribution for all cases might not be fully justified as each group may have a unique sexual partnership pattern. Besides, sexual distributions have been reported to typically have heavy-tailed or power-law tailed property in the large body of literatures (e.g., Schneeberger et al. https://doi.org/10.1097/00007435-200406000-00012, Endo et al. https://doi.org/10.1126/science.add4507, Ito et al. https://doi.org/10.1371/journal.pone.0221520). These studies suggest that a negative binomial may not fully capture the (extreme) level of heterogeneity in sexual partners. I’d like the authors to reanalyse all the groups, using heavy-tailed distributions such as Pareto or Weibull distributions, visually inspect the fit, and/or select a best-fit model for each group by BIC. 

We agree on the value of trying to recognise this problem, as well as how excluding the zero-partner count persons could be handled in sensitivity analysis about the apparent relationship between age and partner counts. Therefore the revised article reports on two model approaches : negative binomial regression on full data by partnership type, as well as applying Weibull distribution to non-zero respondents, with several expressions of the age variable which is still our primary interest.

We selected best model by minimising BIC as suggested. The stata scripts to generate the models and figures are included in supporting information.

6. What type of distributions did the author employ in the regression models with which they tried such as hurdle regression, zero-inflated models, etc (described around P6L186)? Did the authors only use a negative binomial distribution to the other models?

On that note, I do not generally agree with the “data not shown” practice the authors use here, because the statements cannot be interpreted without them along with full description of the methods. Please include in the Supplementary file.

With hurdle regression we tried zero-truncated negbin models, but these failed to converge in latest iterations so not mentioned here.

We want to describe and implement well the models that we do describe; our objective was not to exhaustively try every model form or to identify a best form, but rather to focus on apparent relationship between age & partner count.

We do now fully show (including supplemental material) the two alternative model results and 3 age-variable expressions

7. Regarding the sentence, “The statistical model form reported here (negative binomial) was preferred because the outputs could be easily reported and replicated, supporting development of realistic mpox and sexually transmitted infection models.” at P7L188-191, I think this does not support their model choice at all since there is no statistical justification matched with data. I’m also not sure about what the authors imply by “because the outputs could be easily reported and replicated, supporting development of realistic mpox and sexually transmitted infection models”. 

As mentioned above, my understanding is that sexual partner data have been rarely characterized by negative binomial distributions

Our aim was not to exhaustively explore the model relationship forms, but rather to focus on the age relationship, with some sensitivity analysis addressing the consistency of that relationship. Hopefully, with clarified aim and objectives and multiple models tried, the analysis approach will now seem more justified.

We encourage readers to use the data to their own purposes to explore other research questions and indeed other relationship forms.

We know that some published models have used negbin distribution for sex partner counts:

doi.org/10.1093/ofid/ofac274

doi.org/10.1089/apc.2020.0151

We weren’t in a position to undertake a systematic review to see if negbin distrbtn was rare. We agree with reviewer that it is useful to see if the apparent variations in relationship between age and partner count, especially for MSM, holds up with diverse model forms/distributions for predicting partner count, and now describe methods in context of that strategy.

8. Although the authors repeatedly highlighted their findings are useful in modelling sexually-associated transmissions, there is no description as to how it would be useful. I would be interested in how the authors’ findings could be plugged into disease models. I am not sure if data-driven network modelling for STIs allowing for age-dependent heterogeneity in sexual contact networks is possible as the authors suggest when…

Simply put, many prior models aged MSMs out of the at-risk group at a rigid age point, eg 39 or 65; our data suggest that is unjustified. Moreover, our data suggest that middle aged MSMs may tend to have the highest partner counts.

We hope that changes we made to the aims and objectives in the Introduction now make it much clearer that our focus is on how age and likely partner counts varies with age, it seems unjustified to assume that partner count is about the same from age 16-65 and then abruptly goes to zero. 

We added this sentence to our Limitations section:

“We haven’t tested in transmission models and an exemplar population if varying expected partner count by age makes much difference to outcomes such as total persons made ill or relative effectiveness of possible disease prevention strategies.”

…there is little empirical data on age assortativity in sexual partner formation. 

Age dissassortiveness is an active consideration of many studies about disease transmission among both heterosexuals and MSMs, which often draw on survey data of actual people and use this information in modelled exemplar populations :, eg

doi.org/10.3390/ijerph16091592

doi.org/10.1089/aid.2018.0236

DOI: 10.1097/QAI.0000000000002305

doi. org/10.1136/bmjopen-2020- 039896

https://sti.bmj.com/content/96/1/62.abstract

9. Related to #6; Although the authors said they decided not to present other model approaches (P6L186), I encourage them to show the comparison of all of the fitting including model selection to justify their choice of the final model.

We have declined to show all possible model forms that we explored previously because our aims were not an exhaustive exploration, but rather to explore if the relationship between partner count is likely to be static with age (it’s not) or if there was a sharp decline/change for partner counts at any upper age threshold : we could not observe that sharp decline or a static age-partner count relationship in our data so feel that we have demonstrated that these assumptions (often used in other models) were not supported by our data

10. It would be very informative to show log-log plots for the sexual partnership distributions. This would make readers easily understand how each distribution is skewed.

Partly done, we didn’t know what variable would be log-transformed on the x-axis. We do understand the value of histograms to show the skew, so we added those instead to supplemental file, by partnership type and stratified by 3 age groups (9 plots in new Figure)

11. According to P8L211-212, 28% of MSW are also MSM in the data, which is fairly high compared to the actual proportion of the overlap. This is not surprising because MSM were oversampled due to the nature of the surveys except for Savanta GP, but for this reason I am not sure how we should interpret this 28% figure. I suggest the authors consider alternative ways to characterize the overlap between MSW and MSM.

No action taken.

We chose to focus on the relationship between age and partner counts for the three most common partnership types: MSM, MSW and WSM. We do provide data to enable others interested in WSM&W or MSM&W (etc) or differences between MSoM and MSM&W etc. to undertake their own analyses.

We weren’t entirely sure what characterize the overlap meant. The survey under sampled heterosexual males who are also MSM. We would prefer not to imply we can say something confident about MSW who are also MSM and their likely partner counts. Sample size also means that it could be tricky to create an adequately powered model with many confounders. But ultimately, we thought that exploring less common partnership types was too far outside the scope of our original aims and what conclusions the data could support.

We supply the data and encourage readers to use it to explore other research questions.

12. The authors calculated median of partner counts in each cohort but, as I pointed out in the comment #3, the authors may wish to address the issue of people with zero partner counts when analyzing them (because the median would be sensitive to handling of these people). Probably excluding people with zero partner count would be enough for the purpose of quick comparison. 

We now test and report an alternative model approach that addresses this issue (Weibull regression)

13. In the discussion section (P9L247-P10L265), the authors discussed that the drastic decline in mpox cases would be primarily attributed to behavioural changes citing some papers. However, there is another key factor that would play significant roles in shaping mpox epidemic sizes: that is, depletion of susceptibles effect over heavy-tailed sexual contact networks (Murayama et al. https://doi.org/10.1093/infdis/jiad254; Xiridou et al. https://doi.org/10.1101/2023.01.31.23285294). These studies suggest that accumulation of infection-derived immunity in heavy-tailed sexual networks can dramatically lower the herd immunity threshold and final sizes, and the observed decline in cases may not be primarily attributable to behavioural changes or interventions. I’d like more in-depth discussion in the manuscript that touches on the depletion of susceptibles effect.

No additional discussion added

The survey data provide a snapshot of sexual behaviour; those are the data we have, analysed and are most justified to comment on.

We haven’t introduced the topic of depletion of susceptibles because those data / that hypothesis are based on modelling by others (with caveat and limitations we don’t understand) and that we can’t adequately describe in a brief format suitable for inclusion in the Discussion.

As a side note, ref 15 also discussed this effect as “the biggest factor”. So, it’d be inappropriate to say “early opinion was that decline in the outbreak was more due to behaviour change (15)”, citing ref 15. 

We rephrased our statement to say that behaviour choices were likely important, as below

“early opinion was behaviour change may have been an important reason for decline of the outbreak [15]. …”

14. I do not think discussing the impact of behavioural changes citing ref 14 without noting its limitations is a good idea. Ref 14 did not well inform the impact of behavioural changes on the transmission dynamics as they did not use quantitative measures, alongside that it is unclear from the study whether the results are representative of people who have many partners and are thus playing roles in mpox transmission.

No action taken

This is a fair point but the same and many other potential biases (samples of convenience, recruited on the Internet) apply to most of our own survey data and indeed most modern cross sectional surveys, which we acknowledge in the Methods and Discussion and we believe will be a priori known by readers of this journal. What is not clear that these biases have distorted is the underlying general relationship between age and partner count, allowing for variations by partnership type and age subgroups, so hopefully our analysis about that relationship will sustain further scrutiny and accord with analyses of other large samples in future.

15. Please discuss the strength of the study more convincingly than the authors currently state at P11L304-306. It is indeed a good point that the existing mpox modeling study (ref 2) relied on relatively limited MSM samples, but the authors did not clarify what changes their new data could bring to such studies. For example, the authors’ data indeed has 4x sample size than the data used in ref 2, but is it enough to significantly improve the results of that study (yes confidence intervals may be slightly narrower but is it just that?)?

We hope that with more clearly stated aim and objectives, the statements in discussion about why a more variable relationship between age & partner count is worth considering, are now more convincing. We acknowledge the limitation that we haven’t tested the difference in our own transmission models. We believe that such testing would need to be done to answer the reviewer’s question, and ideally done with a variety of modelling approaches and research questions in order to say definitive things.

I would be convinced if the authors’ data can provide detailed information at the tail part of the sexual partner distributions that had not been available with small sample data. Other aspects of the authors’ data, including sub-samples and the presence of active 65+ individuals, would also be potentially useful in modelling …

It is not clear that we have that much more data about the heavy tail (we haven’t done an exhaustive comparison with other datasets to know), but we do supply the data and explicitly encourage readers to use the data in other analyses to provide more information in their own analyses about heavy-tailed networks or other topics.

Meanwhile, it should also be noted that bias in sampling may undermine the strength of a large sample size. Savanta may be representative as they used quota sampling method, but other cohorts, in particular Grindr, are not. Then the sample size in Savanta becomes 369 (=161+208) and it is almost the same order (a few hundreds the authors said to ref 2 at P11L306). 

Bias in sampling strategy (which we mention repeatedly) is a reason why we haven’t undertaken exhaustive sub group analyses : we aren’t sure what the results would mean. We prefer subgroup analysis only with the larger (> 1000) and most real-world- relevant partnership (most common) types. 

Rather than putting too much emphasis on the sample size, I think it’d be better to put more focus on their detailed information about each data (represented in Table S1-S9), which would be beneficial for those who attempt to construct an age-specific network model. 

We slightly disagree with the referee, In that we argue that The large sample is what enables the rich data to have plausible generalisability. We supply the data and explicitly encourage readers to use the data to test generalisability or predict likely behaviour

Minor comments 

1. The authors often described MSM/MSW/WSM as MSMs/MSWs/WSMs, but “s” should be deleted.

We deleted the s in all cases

2. I recommend the authors make the overall writing more formal, objective, and quantitative throughout. There are a number of instances where the text is rather casual, empty, or vague . I present some examples below:

“We wanted to-” at P2L56,

“Grindr respondents were somewhat younger-“ at P7L195,

“it is quite possible-” at P9L248,

etc. 

We reviewed the manuscript for informal language. Wanted was changed to explored. We removed the word somewhat. We removed the word quite.

3. Typo: “low frequency sexual contact” at P4L122 -> “low frequency of sexual contact” 

“of” was inserted

4. Ref 14 is a summary article based on the original research paper. Please cite this original paper instead (http://dx.doi.org/10.15585/mmwr.mm7135e1).

This reference was replaced with the one indicated

5. In light of the reproducibility of this study and also for the study to make an impact on future research, I hugely encourage the authors to share the data and stata code they made on a platform such as GitHub repository. It is stated that the data will be shared along with another paper but this only makes sense if that paper is published before this paper is published (and I am not fully sure of the point of safeguarding the data when the preprint is already public…?)

The stata code is now included with each fully reported model in the supporting information

The stata code to generate the figures is also included 

6. Please do not use abbreviations in Figures (e.g. ptnrs to partners).

The figures no longer have abbreviations

---

## [Editor Report · Decision Letter 1]

21 Aug 2023

The relationship between age and sex partner counts during the mpox outbreak in the UK, 2022

PONE-D-23-04595R1

Dear Dr. Brainard,

We’re pleased to inform you that your manuscript has been judged scientifically suitable for publication and will be formally accepted for publication once it meets all outstanding technical requirements.

Kind regards,

Soham Bandyopadhyay

Academic Editor

PLOS ONE

---

## [Editor Report · Acceptance letter]

29 Aug 2023

PONE-D-23-04595R1 

The relationship between age and sex partner counts during the mpox outbreak in the UK, 2022 

Dear Dr. Brainard:

I'm pleased to inform you that your manuscript has been deemed suitable for publication in PLOS ONE. Congratulations! Your manuscript is now with our production department. 

Kind regards, 

on behalf of

Dr. Soham Bandyopadhyay 

Academic Editor

PLOS ONE